# Unplanned Weight Loss and Sarcopenia Across Body Mass Index Categories in Nursing Homes—A Cross-Sectional Study

**DOI:** 10.3390/nu17010171

**Published:** 2025-01-02

**Authors:** Laura Klaassen, Tenna Christoffersen, Margit Dall Aaslyng, Inge Tetens

**Affiliations:** 1LUMC Center of Medicine for Older People, Leiden University, 2333 ZA Leiden, The Netherlands; l.klaassen@umail.leidenuniv.nl; 2Department of Nutrition, Exercise and Sports, Faculty of Science, University of Copenhagen, 1870 Frederiksberg, Denmark; tch@pha.dk (T.C.); ite@nexs.ku.dk (I.T.); 3Nutrition and Health, University College Absalon, 4200 Slagelse, Denmark

**Keywords:** community care setting, elderly, obesity, muscle loss, nutritional issues

## Abstract

**Background**: Nutritional risks in older adults, such as malnutrition and sarcopenia, are often underdiagnosed. Screening practices frequently rely on Unplanned Weight Loss (UPWL), potentially overlooking at-risk individuals. This study aims to assess the prevalence of nutritional risk, identified by UPWL and sarcopenia, across different body mass index categories in a nursing home (NH) population. **Methods**: Cross-sectional data were collected from an NH in a Danish municipality, including those of all self-reliant participants who consented and excluding those of terminally ill older adults. Data on age, sex, height, weight, and chronic diseases were extracted from medical records. Nutritional risk was assessed using two markers: a UPWL of ≥1 kg during the last six months and muscle strength via a modified 30 s chair stand test as a marker of sarcopenia. An ANOVA and Fisher’s Exact Test were used to assess differences, followed by a post hoc Tukey test. **Results**: In our study of older adults (*n* = 93, mean age 83.2 ± 9.12 years, 63% female), 17 individuals (19%) had UPWL, and 27 (29%) had sarcopenia. Among those with obesity, twelve (48%) had sarcopenia, but only two (8%) had UPWL. In contrast, seven (21%) of those with normal weight had sarcopenia, while eleven (33%) experienced UPWL. **Conclusions**: Older adults in NHs are at nutritional risk, but the prevalence varies significantly depending on whether UPWL or sarcopenia markers are applied for categorization. Sarcopenia prevalence was the highest in the obesity group, suggesting a need for integrating muscle strength or quantity assessments into community care to identify older adults at nutritional risk better.

## 1. Introduction

Nutritional risk is the likelihood of developing health issues due to inadequate or imbalanced nutrition [1]. Nutritional risk screening is a rapid method for identifying older adults needing nutritional care [1]. However, nutritional syndromes such as malnutrition and sarcopenia are often underdiagnosed [2,3,4] and commonly overlap in nursing home (NH) residents [1,3]. Malnutrition and sarcopenia are linked to poor nutrition, aging, and the loss of muscle mass and function, leading to inactivity, disabilities, reduced Quality of Life (QoL), increased risk of comorbidity and mortality, and higher healthcare costs [5,6,7]. Although malnutrition and sarcopenia syndromes are preventable and treatable if detected early [2,3,4,7], NHs mainly assess Unplanned Weight Loss (UPWL) as a marker of nutritional risk [1]. Standard screening for nutritional risk includes an assessment of UPWL, followed by an assessment of reduced food intake, low body mass index (BMI), and disease activity [1].

In Denmark, the main criterion for initial nutritional risk screening is a UPWL of ≥1 kg [8]. However, assessing muscle mass, crucial for diagnosing sarcopenia [9,10], is uncommon in community care settings like NHs [11]. Danish practice may overlook some older adults at risk of malnutrition and/or sarcopenia, since not all older adults at nutritional risk experience UPWL. By focusing on weight loss and low BMI, healthcare practitioners might overlook individuals with stable or higher body weight who are still at nutritional risk [1].

Older adults who are overweight or obese may also suffer from nutritional syndromes, even if they do not experience UPWL or have a low BMI. Consequently, many older adults in the early stages of nutritional syndromes remain unidentified. This is a concern since malnutrition and sarcopenia can synergistically worsen each other and exacerbate other health issues, making identification and intervention crucial [2]. Therefore, relying mainly on UPWL might not be the best way of assessing nutritional risk, since it can fail to identify at-risk individuals who have not lost weight or are overweight or obese. Thus, we hypothesize that some older adults at nutritional risk may be overlooked if UPWL is used as the main risk marker.

In this study, we aim to assess and compare the prevalence of nutritional risk in older adults, identified by UPWL and sarcopenia, across different BMI categories in an NH population.

## 2. Materials and Methods

This cross-sectional study was conducted in six NHs in Odsherred Municipality, Region Zealand, Denmark, in March 2023 as part of the ‘Interventions focusing on nutrition and physical activities for overweight and obese older adults in NHs’ “IFEBO” study [Clinical. Trial: NCT05804019]. In Denmark, the state provides personal and practical care, but residents must pay for housing, meals, medications, and services such as physiotherapy. This project adhered to the principles outlined in the Declaration of Helsinki II, as well as the guidelines set forth by the Regional Ethics Committee [EMN-2021-07672], and was approved by the Ethical Committee of Copenhagen University [J.no.: 504-0316/22-5000].

### 2.1. Data Sources

Older adults were recruited based on the following inclusion criteria: self-reliance and willingness to provide informed consent. Terminally ill older adults were excluded by the NH management team to ensure participant ethics and suitability. Data were collected at this study’s baseline, acknowledging that not all participants completed every measurement.

The primary investigators (TC and LK) collected data from medical records including age, sex, height, weight, and chronic diseases. BMI was calculated by dividing weight (kg) by height (m) squared (kg/m^2^). Participants were classified according to BMI categories as underweight (˂18.5 kg/m^2^), normal weight (≥18.5 kg/m^2^ ˂24.9 kg/m^2^), overweight (≥25 kg/m^2^ ˂29.9 kg/m^2^), and obese (≥30 kg/m^2^), in alignment with guidelines from the World Health Organization (WHO) and the European Society of Clinical Nutrition and Metabolism (ESPEN) [1,12].

Monthly weight change data recorded by care staff in the participants’ medical records were used to assess nutritional risk. In Denmark, a UPWL of ≥1 kg or several consecutive weight losses totaling more than 1 kg in the last six months are the initial criteria for assessing nutritional risk in older adults [8].

### 2.2. Diagnostic Criteria and Assessment of Sarcopenia

The diagnosis of sarcopenia was assessed using the European Working Group on Sarcopenia in Older People (EWGSOP2) revised criteria and cut-off points, which include four steps: case-finding, assessing probable sarcopenia through low muscle strength, confirming the diagnosis via low muscle mass or quality, and grading severity by physical performance [10]. Case-finding was applied to all participants based on the clinical suspicion of sarcopenia [10] upon their admission to NHs due to their functional disabilities and lack of independence.

Probable sarcopenia was assessed by using a 30 s chair stand test (30 s CST) [10]. A 30 s CST result of ≤5 stands suggests low muscle strength, indicative of sarcopenia [13]. The diagnosis of sarcopenia was confirmed when these probable criteria were met, along with calf circumference (CC) measurements of <31 cm for both sexes, adjusted for BMI [10]. An adjusted CC was calculated by adding 4 cm for a BMI of <18.5 kg/m^2^ or subtracting 3, 7, or 12 cm for BMIs of 25–29, 30–39, and ≥40 kg/m^2^, respectively [14]. All measurements were performed according to standard procedures [15,16].

### 2.3. Statistical Analyses

Descriptive statistics were used to summarize the data from the participants. Weight change included all weight changes, planned or unplanned, gain or loss, during the last six months, whereas UPWL excluded participants with planned weight loss or a weight loss of <1 kg. An Analysis of Variance (ANOVA) or Fisher’s Exact Test was used to assess potential differences in categories. Subsequently, post hoc Tukey tests were utilized to compare BMI categories for significant differences. Statistical analyses were conducted using IBM^®^ SPSS version 27.

## 3. Results

A total of 93 out of 235 residents were enrolled in this study. Most of the participants were female (63%) with an average age of (mean ± SD) 83.2 ± 9.12 years (Table 1). Approximately one-third of the participants were categorized as normal weight (36%), followed by those who were categorized as overweight or obese (30% and 29%, respectively). Regarding health conditions, non-communicable diseases (NCDs) were highly prevalent (88%), with cardiovascular disease (CVD) being the most common (62%). The prevalence of comorbidity was high (95%). When comparing BMI categories, participants with obesity were shown to have experienced a significantly greater weight change in the last six months compared with participants with normal weight (*p* = 0.007).

UPWL was observed among participants with normal weight, overweight, and obesity (Table 2). The highest prevalence of UPWL was observed in the normal weight group (mean: −3.32 kg; SD: 1.74), followed by those with overweight (mean: −5.18 kg; SD: 3.82) and obesity (mean: −5.75 kg; SD: 2.19) (*p* = 0.26). UPWL showed a significant inverse correlation with BMI categories (*p* = 0.03). By contrast, sarcopenia prevalence showed a significant positive correlation (*p* = 0.04). Neither UPWL nor sarcopenia was identified among participants categorized as underweight.

## 4. Discussion

The main finding of our study was that when focusing primarily on UPWL in NHs, the nutritional risk may be underscored, especially when considering the BMI category of obese older subjects. This suggests that the identification of nutritional risks across BMI categories may be an important measure in the nutritional screening of older adults in NHs. UPWL was found to be more common in the normal weight group than in the obesity group. Conversely, when using the sarcopenia diagnosis, the prevalence was higher in the obesity group.

In a comparative cross-sectional study of 176 Dutch NH residents, 16% experienced a UPWL of either ≥3 kg during the last month or ≥6 kg over the last six months (8% + 8%, respectively) [17], which is similar to our study where 19% experienced UPWL. However, in our study, the criteria were quite different (BW loss ≥ 1 kg months) [8] from those of the Dutch study, where the weight loss was considerably higher (≥6 kg within the last six months or ≥3 kg during the last month) [18]. This difference calls for the harmonization of nationally used tools to allow for international comparisons.

In a Swedish cohort study of 47,686 NH residents, nutritional risk was assessed using the Mini Nutritional Assessment-Short Form (MNA-SF). This tool includes UPWL during the past three months, specific BMI thresholds, low food intake, low mobility, psychological stress or acute disease during the past three months, and neuropsychological problems such as dementia or depression [19]. The study describes a prevalence of 45% of residents who were at nutritional risk, more than twice that identified by UPWL alone in our study. Another cohort, including 485 NH residents, finds that 75% were at nutritional risk, according to the MNA-SF [20].

However, the MNA-SF encompasses more factors than just UPWL, and the study did not report the prevalence of each criterion. In addition, the researchers found that 8% of the participants in the cohort study were categorized as underweight, 45% as normal weight, 31% as overweight, and 16% as obese [19]. By contrast, our study reveals higher obesity rates and a lower prevalence of underweight, normal weight, and overweight. Notably, the Swedish cohort study showed an inverse association between obesity and mortality, regardless of nutritional status, as assessed by the MNA-SF [19]. This observation calls for the need to include several criteria as part of the screening tool if the nutritional risk should be related to mortality or other outcomes.

The American National Health and Nutrition Examination Survey (NHANES) cross-sectional study reports BMI categories among US NH residents (n = 7261) as 1% underweight, 25% normal weight, 37% overweight, and 37% obese [21]. Compared with our study, the prevalence of underweight and normal weight is lower, while overweight and obesity rates are higher [21]. The NHANES study found that comorbidity is associated with obesity and is linked to functional limitations, which may indicate low muscle strength. However, it did not assess nutritional risk or mortality [21]. A large cross-sectional study from Europe involving 8938 community-dwelling participants across 21 countries reports BMI categories as follows: 0% underweight, 37% normal weight, 42% overweight, and 21% obese [22]. In our study, we found a higher proportion of obese residents compared to overweight residents than what was reported in the European study [22]. This difference may suggest that older adults with obesity are more likely to be admitted to nursing homes, potentially due to the age-related loss of muscle mass combined with obesity.

Earlier definitions of sarcopenia were characterized by the loss of muscle mass, but now, more focus is placed on the loss of muscle function, especially muscle strength [9].

Our findings show a sarcopenia prevalence of 29%, which is consistent with similar international studies, which report a range of 18% to 29% among older adults in NH [3,23,24]. Variations in sarcopenia prevalence across studies can partly be explained by different muscle assessment methods. In other studies, muscle strength was measured using Hand Grip Strength (HGS) and/or the 30 s CST, while muscle mass was evaluated using Dual-Energy X-ray Absorptiometry (DXA), Bioelectrical Impedance Analysis (BIA), or CC [3,23,24].

This study has some strengths and several limitations. Its cross-sectional design does not capture changes in nutritional syndromes over time [1]. However, our design allows for the timely assessment and updating of nutritional risk prevalence across BMI categories. A strength of this study is the employment of methods aligned with EWGSOP2 and ESPEN guidelines to assess nutritional syndromes. Despite the limited sample size, this study employed robust and validated measurements.

## 5. Conclusions

Our study found that 19% and 29% of older adults in nursing homes are at nutritional risk when identified by Unplanned Weight Loss (UPWL) and sarcopenia diagnosis, respectively. The highest UPWL prevalence was observed among older adults categorized as normal weight (≥18.5 ˂24.9 kg/m^2^). At the same time, older adults with obesity (≥30 kg/m^2^) are at nutritional risk, with approximately half being diagnosed with sarcopenia despite experiencing less frequent UPWLs. Relying mainly on UPWL to identify nutritional risk in nursing home populations may be inadequate and could potentially result in at-risk older adults being overlooked. Therefore, integrating muscle strength and/or quantity assessments can provide a comprehensive identification of nutritional risk in community care settings.

## Figures and Tables

**Table 1 nutrients-17-00171-t001:** Characteristics of participating older adults in NHs classified by BMI categories (mean, standard deviation (SD), or prevalence n (%)).

Variables	Total *n* = 93	Underweight *n* = 5	Normal Weight *n* = 33	Overweight *n* = 28	Obese *n* = 27	*p*-Value
Sex, female, *n* (%) Age, years, mean (SD)	59 (63.4)	4 (80.0)	21 (63.6)	13 (46.4)	21 (77.8)	0.1
	83.2 (9.1)	89.2 (8.3)	84.4 (9.7)	82.5 (8.3)	81.3 (9.1)	0.3
Body composition Body weight, kg, mean (SD) CC adjusted for BMI, cm, mean (SD)	75.7 (16.9)	44.8 (6.1)	64.0 (9.0)	80.1 (11.0)	90.9 (12.9)	<0.0001
	32.5 (3.9)	32.4 (1.4)	32.8 (3.4)	33.2 (4.4)	31.2 (4.2)	0.2
Body weight change during last 6 months						
Average change, kg, mean (SD)	+2.4 (6.8)	−0.6 ^b^ (1.0)	+0.6 ^b^ (5.8)	+2.3 ^ab^ (6.8)	+5.3 ^a^ (7.8)	<0.05
Participants losing weight, *n* (%)	28 (30.1)	2 (40.0)	13 (39.4)	9 (32.1)	4 (14.8)	<0.05
Weight loss, kg, mean (SD)	−4.3 (4.1)	−1.5 (1.1)	−4.3 (4.5)	−5.2 (4.5)	−4.6 (9.9)	0.8
Participants gaining weight, *n* (%)	58 (62.4)	1 (20.0)	18 (54.6)	18 (64.3)	21 (77.8)	<0.05
Weight gain, kg, mean (SD)	+6.0 (5.6)	+0.1 (-) *	+4.1 (4.3)	+6.2 (4.2)	+7.6 (7.0)	0.2
Diseases						
NCDs, *n* (%)	82 (88.2)	4 (80.0)	27 (81.8)	26 (92.9)	25 (92.6)	0.5
CVD, *n* (%)	58 (62.4)	4 (80.0)	19 (57.6)	17 (60.7)	18 (66.7)	0.6
T2D, *n* (%)	24 (25.8)	0 (0)	6 (18.2)	8 (28.6)	10 (37.0)	0.8
Cancer, *n* (%)	13 (14.0)	0 (0)	5 (15.2)	6 (21.4)	2 (7.4)	0.2
CRDs, *n* (%)	11 (11.8)	0 (0)	6 (18.2)	3 (10.7)	2 (7.4)	0.5
Comorbidity, *n* (%)	88 (94.6)	5 (100)	29 (87.9)	27 (96.4)	27 (100)	0.2
Muscle Strength **						
30 s CST, mean (SD)	2.4 (3.2)	1.4 (3.1)	2.6 (3.6)	2.9 (3.3)	1.9 (2.6)	0.6

Abbreviations: 30 s CST: 30 s chair stand test; BMI: body mass index; CC: calf circumference; n: number of subjects; SD: standard deviation; NCDs: non-communicable diseases; CVD: cardiovascular disease; T2D: type 2 diabetes; CRDs: chronic respiratory diseases; NH: nursing home; * SD is not calculable due to a single observation. ** 2 NA in BMI group: obesity. ^a^ and ^b^ different letters in the same line indicate significant differences (*p* < 0.05).

**Table 2 nutrients-17-00171-t002:** The prevalence of Unplanned Weight Loss ≥1 kg (UPWL) and sarcopenia diagnosis classified by BMI categories (prevalence n (%)).

	Total *n* = 91	Underweight *n* = 5	Normal Weight *n* = 33	Overweight *n* = 28	Obese * *n* = 25	*p*-Value
Unplanned Weight Loss ≥1 kg in 6 months, *n* (%) Sarcopenia Diagnosis, *n* (%)	17 (18.7)	0 (0)	11 (33.3)	4 (14.3)	2 (8.0)	0.03
	27 (29.0)	0 (0) ^b^	7 (21.2) ^b^	8 (28.6) ^ab^	12 (48.0) ^a^	0.04

UPWL excluded participants with planned weight loss or weight loss <1 kg. *n*: number of subjects; * 2 NA in BMI group: obesity. ^a^ and ^b^ different letters in the same line indicate significant differences (*p* < 0.05).

## Data Availability

Data are unavailable due to privacy and ethical restrictions.

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
