# Peer review of "Unplanned Weight Loss and Sarcopenia Across Body Mass Index Categories in Nursing Homes—A Cross-Sectional Study"

_nutrients, 2025, doi:10.3390/nu17010171_

Round 1
Reviewer 1 Report
Comments and Suggestions for Authors
This research delves into the nutritional vulnerabilities of older adults in nursing homes, specifically examining how risks are detected through indicators such as Unplanned Weight Loss (UPWL) and sarcopenia (muscle weakness) across various body mass index (BMI) categories. The findings offer intriguing insights for health policymakers and underscore the importance of this research for the health and well-being of the elderly.
However, I have a primary suggestion regarding this article. The data is mainly presented in Tables 1 and 2, with Table 2 being quite clear. However, Table 1 is not as easy to comprehend, particularly in the sections on 'Body Weight Change in 6 months' and 'Muscle Strength'. I recommend that the authors provide additional explanations and clarifications for the relevant data.
Author Response
Comment 1:
This research delves into the nutritional vulnerabilities of older adults in nursing homes, specifically examining how risks are detected through indicators such as Unplanned Weight Loss (UPWL) and sarcopenia (muscle weakness) across various body mass index (BMI) categories. The findings offer intriguing insights for health policymakers and underscore the importance of this research for the health and well-being of the elderly.
Response 1: Thank you for your thoughtful and encouraging feedback. We truly appreciate your recognition of the importance of addressing nutritional vulnerabilities in older adults.
Comment 2: The data is mainly presented in Tables 1 and 2, with Table 2 being quite clear. However, Table 1 is not as easy to comprehend, particularly in the sections on 'Body Weight Change in 6 months' and 'Muscle Strength'. I recommend that the authors provide additional explanations and clarifications for the relevant data.
Response 2: We agree. Table 1 provides additional explanations. We elaborate on the section labels. Furthermore, as suggested by reviewer no. 2, we also revised the number of decimals for better interpretation.
Reviewer 2 Report
Comments and Suggestions for Authors
title could be shortened
line 42 - why this space
line 68 - it would be interesting to explain if the national health insurance covers the cost of all nursing homes or patients pay themselves
table 1 - you wrote 0.04 and under that less than 0.05, pick one type of data presentation. Also some numbers are presented with 1 decimal number and some with two
line 161 - it would be interesting to compare national criteria with those from relevant global authorities
line 163 - I believe you have NH abbreviation introduced, check whole manuscript or delete this abbreviation from abstract
you only have 22 references, since body of literature in older population and malnutrition is rising, I believe you could include more recently published relevant studies and discuss your results in light of previously conducted research
This is interesting and well conducted study. Methods are nicely described and results presented. I woull add more up to date information to make the manuscript more interesting to future readers. I believe it does add to the exsisting body of literature
Author Response
Comment 1: title could be shortened
Response 1: We agree. We shortened the title to: Unplanned Weight Loss and Sarcopenia Across Body Mass Index Categories in Nursing Homes – A Cross-Sectional Study.
Comment 2: line 42 - why this space.
Response 2: Space deleted.
Comment 3: line 68 - it would be interesting to explain if the national health insurance covers the cost of all nursing homes or if patients pay themselves.
Response 3: We added the information in line 74.
Comment 4: table 1 - you wrote 0.04 and under that less than 0.05, pick one type of data presentation. Also some numbers are presented with 1 decimal number and some with two.
Response 4: The numbers in Table 1 has been revised applying one decimal.
Comment 5: line 161 - it would be interesting to compare national criteria with those from relevant global authorities
Response 5: We appreciate the reviewer’s suggestion to compare our results with the criteria established by international bodies; however, due to constraints in scope and the specific focus of our study, we regret that it is not feasible to incorporate this comparison at this time.
Comment 6: line 163 - I believe you have NH abbreviation introduced, check whole manuscript or delete this abbreviation from abstract.
Response 6: The whole manuscript is updated using the NH abbreviation.
Comment 7: you only have 22 references, since body of literature in older population and malnutrition is rising, I believe you could include more recently published relevant studies and discuss your results in light of previously conducted research.
Response 7: We’ve added two recent articles (references 20 and 22, published in 2021) to improve the manuscript’s relevance.
Comment 8: This is an interesting and well-conducted study. Methods are nicely described and results presented. I would add more up-to-date information to make the manuscript more interesting to future readers. I believe it does add to the existing body of literature.
Response 8: Thank you for your positive and constructive feedback. We appreciate your kind words about the study's methodology and presentation. We’ve added two articles to the discussion (reference 20 and 22).